# Phytochemical Characterization and Assessment of the Wound Healing Properties of Three Eurasian Propolis

**DOI:** 10.3390/ph17111412

**Published:** 2024-10-22

**Authors:** Elisabetta Miraldi, Alessandro Giordano, Giorgio Cappellucci, Federica Vaccaro, Marco Biagi, Giulia Baini

**Affiliations:** 1Department of Physical Sciences, Earth and Environment, University of Siena, 53100 Siena, Italy; alessandr.giordano@student.unisi.it (A.G.); giorgi.cappellucci@unisi.it (G.C.); f.vaccaro@student.unisi.it (F.V.); giulia.baini2@unisi.it (G.B.); 2Department of Food and Drug, University of Parma, 43124 Parma, Italy; marco.biagi@unipr.it

**Keywords:** propolis, Eurasian propolis, chemical characteristics, anti-inflammatory properties, immunomodulatory properties

## Abstract

Objectives: The objective of this study is to evaluate the wound healing potential of Eurasian propolis by analyzing the phytochemical profile and the biological effects of three representative propolis samples. Methods: Specific colorimetric assays were used to estimate the total phenolic and flavonoid contents and the triterpenoids content. Some of the main components of Eurasian propolis (pinocembrin, pinobanksin, CAPE, chrysin and galangin) were analyzed using HPLC-DAD. Scavenging activity and total antioxidant capacity were assessed through DPPH and ORAC assays, respectively. Human keratinocyte, fibroblast, and monocytic cell lines were used for the biological in vitro analyses. The direct wound healing properties were tested through scratching assays and ELISA kits for the assessment of the production of growth factors (FGF-7, Latency Associated Peptide-LAP), while the indirect effects were evaluated through the estimation of the levels of MMP9, IL-1β, IL-8, and TNF-α using ELISA kits together with a cell-free test on the inhibition capacity on collagenases. Network Pharmacology analysis was employed to further explore possible mechanisms of the action of propolis on the healing process. Results: The analyses confirmed the high phenolic content of Eurasian propolis (142.50–211.30 mg GAE/g), dominated by flavonoids (95.50–196.80 mg Galangin Equivalents/g), and terpenes (431.50–650.00 mg β-sitosterol Equivalents/g), while also verifying the significant antioxidant (4.9–8.9 mM/g Trolox Equivalents) and antiradical (DPPH IC_50_ 26.1–54.4 μg/mL) activities. The samples showed indirect wound healing properties by mitigating inflammation and remodeling (reduced IL-1β and MMP9) and potentially modulating the immune response (upregulated IL-8). In vitro studies confirmed these effects, demonstrating decreased MMP9 production and collagenase inhibition when cells were co-treated with propolis and a stressor. Propolis also suppressed IL-1β release in fibroblasts, although its impact on TNF-α was inconclusive. Notably, co-treatment upregulated IL-8 in monocytes, suggesting a potential immunomodulatory role. Conclusions: Eurasian propolis may not directly stimulate cell proliferation during wound healing. Its anti-inflammatory and immunomodulatory properties could indicate an indirect contribution in helping the process.

## 1. Introduction

With its peculiar structure and composition, the skin is an organ that acts both as a barrier and an interface of the human body with the external environment. From blocking outer hazards from entering or affecting the organism to granting humans the ability to sense the world around them and adapt to it, this organ, the largest of the human body, possesses functions and mechanisms essential for life. However, it is necessary to highlight the vulnerability of this tissue to damage. A wound is a damage to the natural structure, integrity and functions of the skin caused by a mechanical, chemical or physical agent. The injury modifies the communication skills within the cutaneous tissue itself and with the surrounding environment. This can potentially expose the organism to infections, loss of mobility, pain and non-healing of the wounds, dramatically affecting the patients’ quality of life [1]. In the four stages of the healing process (haemostasis, inflammation, proliferation, remodeling), a key role is played by several factors released by the cells present and attracted at the site of the injury. These include cytokines involved in the inflammatory process and chemokines for the recall of immunity cells, GFs which stimulate the proliferation of the epithelial and endothelial cells and remodeling enzymes like the MMPs which re-elaborate the ECM after the damage to the skin [2].

Due to the current lack of effective treatment options, part of the research for possible stimulating agents of the wound healing process has shifted to natural products. The choice for this change is justified by the significant pharmacological properties they show, conferred by their components and phytocomplexes [3]. The employment of medicinal herbs and plant-derived substances in wound healing has the aim of reducing the onset of complications such as infection or chronicization of the wound [4]. Undoubtedly, the specific composition of every medicinal plant/natural substance influences the effects on the illness or physiological pathway under study. The response variability, also observable in the same categories of plants/materials, is often due to differences in the quantity of single components, each one of them with distinct levels of bioactivity.

Propolis, a mixture of substances gathered by bees from a variety of plant sources, is one of the most used plant-derived products for topical application since ancient times. Its antimicrobial, anti-inflammatory and antioxidant properties make it a perfect example of a natural substance useful for enhancing the wound healing process. [5]

Given the considerable weight of the local flora in propolis composition, it is possible to classify distinct types of propolis based on the geographical area. Examples are the Eurasian propolis from temperate zones rich in flavonoids and phenolic acid esters, essential molecules for the properties of the mixture. Yet, it is not uncommon to observe differences in the amounts of chemicals in propolis from the same class too, such as polyphenols and flavonoids, which, as the literature shows, can range from 3 up to 40% (*w*/*w*) in Eurasian propolis. There are some chemical markers, such as pinocembrin, galangin, chrysin and CAPE, that are generally present, albeit in different amounts, depending on parameters such as the area and period of production. The variability in composition of the subsequent formulation based on the extraction method used should also be noted [6,7,8,9,10,11,12,13].

In this work the differences between three Eurasian propolis will be evaluated chemically and biologically, focusing on the different stimulation of the wound-healing process on both keratinocytes and fibroblast. The objective is to highlight the importance of the composition of various propolis formulations on the healing of injuries, while also evaluating the influence of extraction methods.

## 2. Results

### 2.1. Phytochemical Analysis

The results of the Folin–Ciocâlteu assay, expressed as GAE ± SD, were 211.30 ± 11.70 mg/g (Sample 1), 142.50 ± 5.70 mg/g (Sample 2) and 185.00 ± 9.00 mg/g (Sample 3) (Table 1). The obtained data have confirmed that phenolic molecules are one of the main components of propolis, especially given the numerous biological properties ascribed to this chemical family, such as antioxidant properties, anti-inflammatory properties and effects on the gut microbiota [14,15,16].

The obtained total flavonoids concentration, expressed as Galangin Equivalents ± SD, were 196.80 ± 3.60 mg/g (Sample 1), 95.50 ± 2.30 mg/g (Sample 2) and 130.90 ± 4.60 mg/g (Sample 3) (Table 2). Although it is not possible to make a direct comparison with the results in Table 1 due to the application of different analysis methodologies, these data confirm, as expressed in the literature, that flavonoids make up the majority of the phenolic component present in propolis. Their presence justifies the numerous biological activities attributed to propolis itself, such as antioxidant, antimicrobial and anti-inflammatory properties [13].

The determined triterpenoid contents, expressed as β-sitosterol Equivalents ± SD, were 611.90 ± 6.00 mg/g (Sample 1), 431.50 ± 3.10 mg/g (Sample 2) and 650.00 ± 6.40 mg/g (Sample 3) (Table 3). The elevated quantities of triterpenes found in the analyzed samples highlight the resinous nature of propolis.

Compared to Samples 1 and 3, Sample 2 exhibited an inferior triterpenoid concentration, probably due to the industrial procedures of the removal of waxes, extraction and purification to which it was subjected.

The HPLC-DAD analysis of the samples produced the following results for the concentrations of the five main molecules of interest, expressed as % *w*/*w* ± SD. Sample 1: 1.46 ± 0.27% (pinobanksin), 8.14 ± 1.40% (pinocembrin), 1.93 ± 0.34% (CAPE), 13.22 ± 2.25% (galangin + chrysin). Sample 2: 0.66 ± 0.08% (pinobanksin), 2.88 ± 0.42% (pinocembrin), <0.01% (CAPE), 8.60 ± 2.76% (galangin + chrysin). Sample 3: 1.29 ± 0.05% (pinobanksin), 3.68 ± 0.81% (pinocembrin), <0.01% (CAPE), 12.68 ± 2.23% (galangin + chrysin) (Table 4).

It is worth noting that CAPE was only detectable in Sample 1, perhaps due to the industrial processes which the other samples were subjected to. The peaks of chrysin and galangin could not be separated and were considered as one. Taking into account the coelution of these two molecules, pinocembrin proved to be the main constituent of the Eurasian propolis under study, followed by chrysin and galangin.

Below are the registered chromatograms (Figure 1). On a qualitative level, it can be noted that chromatograms of Sample 1, 2 and 3 showed a similar peak profile, reflecting the similar composition of the starting material of the samples (raw Eurasian propolis).

### 2.2. Biological Analysis and Network Pharmacology

The results obtained from the DPPH assay (IC_50_ expressed as μg/mL ± SD) are 26.09 ± 1.30 (Sample 1), 54.38 ± 2.72 (Sample 2) and 45.50 ± 2.28 (Sample 3) (Table 5), while those of the ORAC assay, expressed as mM/g ± SD of Trolox Equivalents, are 8.89 ± 1.50 (Sample 1), 4.93 ± 2.21 (Sample 2) and 6.96 ± 2.05 (Sample 3) (Table 6). For each sample, both the radical scavenging capacity and the antioxidant power are in line with the respective concentrations of total phenolics and flavonoids.

The observed cytotoxicity assay outcomes on HFF cells were 100.00 ± 6.23% (Control), 97.50 ± 4.16% (Sample 1), 98.28 ± 5.78% (Sample 2) and 91.17 ± 6.82% (Sample 3) ([Fig pharmaceuticals-17-01412-ch001]), basically showing an absence of toxicity in all propolis samples at the tested concentrations.

The scratch wound healing assays carried out resulted in no improvement of the healing process by the samples (mean relative healing of samples < mean relative healing of Control), with sporadic exceptions that were not significant (example of HaCaT healing in Figure 2). These results suggested that propolis may not have a direct effect on the healing of wounds by stimulating the proliferation or the migration of the epithelial cells.

The study proceeded with the quantification of two of the main GFs responsible for the epithelial cell proliferation and growth: FGF and TGF-β.

The quantification of FGF-7 produced the following relative results: 1.004 ± 0.088 (Control), 0.986 ± 0.054 with *p* > 0.05 (Sample 1), 1.037 ± 0.085 with *p* > 0.05 (Sample 2) and 1.024 ± 0.066 with *p* > 0.05 (Sample 3). Those obtained for LAP are: 1.002 ± 0.070 (Control), 0.989 ± 0.059 with *p* > 0.05 (Sample 1), 1.020 ± 0.046 with *p* > 0.05 (Sample 2) and 1.007 ± 0.051 with *p* > 0.05 (Sample 3) ([Fig pharmaceuticals-17-01412-ch002]).

These null outcomes align with those of the scratch wound healing assays, pointing to the same conclusion: propolis has a healing effect on wounds confirmed by traditional use, but this may not be due to a direct effect on the epithelial cells through the stimulation of their proliferation and/or migration.

The inclusion of the five molecules analyzed and recognized by HPLC-DAD (pinobanksin, pinocembrin, CAPE, galangin, chrysin) as input in SwissTargetPrediction led to an important convergence of prediction towards the MMP enzyme family: in particular, as shown in Table 7, four matches for MMP2 (mean probability 0.1128), four matches for MMP9 (mean probability 0.1128), four matches for MMP12 (mean probability 0.1422) and three matches for MMP13 (mean probability 0.1480) were highlighted.

Although the matching probabilities are significantly lower than the standard references reported by the developers of molecular prediction software (Combined-Score higher than 0.5) [17], this finding was nevertheless taken into account, considering that in the heterogeneity of the propolis phytocomplex, most of the five molecules taken into account and analyzed with SwissTargetPrediction share the MMP target. This suggests a plausible increase in the probability of prediction in reality due to the matrix effect.

The quantification of MMP9 gave the following results: 1.047 ± 0.342 (Control), 2.211 ± 0.413 with *p* < 0.001 (LPS + H_2_O_2_), 0.983 ± 0.267 with *p* < 0.001 (Sample 1 + Stresses), 1.146 ± 0.218 with *p* < 0.001 (Sample 2 + Stresses) and 1.490 ± 0.413 with *p* < 0.01 (Sample 3 + Stresses) ([Fig pharmaceuticals-17-01412-ch003]).

The clear outcomes of this experiment show the possible ability of propolis to modulate one of the main phases of the wound-healing process, the remodeling, by limiting the over-production of MMP9 in response to a damaging stimulus (here both inflammatory and oxidative stress).

Another step was to evaluate propolis direct activity on the ECM-degrading enzymes. The collagenase inhibition assay results (IC_50_ expressed as μg/mL ± SD) are 41.61 ± 2.08 (Sample 1), 113.96 ± 5.70 (Sample 2) and 51.57 ± 2.58 (Sample 3) (Table 8).

Together with the results on MMP9 synthesis, these outcomes point to a probable action of propolis on the whole remodeling phase of wound healing, by simultaneously decreasing the overproduction of MMPs after damage and inhibiting the enzymes already in the site.

To expand the pool of possible targets of propolis involved in the wound-healing process, the research moved on to other in silico target prediction tools.

By inserting twenty-four targets (see Section 4.5) derived from the analysis of the five marker molecules for Eurasian propolis with SwissTargetPrediction, thirty new targets were proposed by GeneRecommender AI engine as output (Figure 3). Among these possible targets, some of them pointed to a possible influence of Eurasian propolis main constituents on both inflammation and immunomodulation.

The possible anti-inflammatory activity during wound healing was firstly evaluated on stimulated fibroblasts.

Quantification of IL-1β produced, as relative results, 1.002 ± 0.064 (Control), 1.133 ± 1.117 with *p* < 0.01 (LPS + H_2_O_2_), 0.973 ± 0.099 with *p* < 0.001 (Sample 1 + Stresses), 1.046 ± 0.041 with *p* < 0.05 (Sample 2 + Stresses) and 0.948 ± 0.056 with *p* < 0.001 (Sample 3 + Stresses) ([Fig pharmaceuticals-17-01412-ch004]).

Even though the stresses did not produce an impressive response on the cells, the co-treatments showed a statistically significant reduction in the production of IL-1β, one of the main pro-inflammatory cytokines.

A great part in the wound-healing process is played by the immune cells already present in the damaged skin or summoned from the bloodstream. In addition to the anti-inflammatory effect of propolis on immunity cells, its immunomodulatory activity was also studied.

The quantification of IL-8 in treated THP-1 gave the following relative outcomes: 1.002 ± 0.080 (Control), 0.982 ± 0.057 with *p* > 0.05 (Sample 1), 0.949 ± 0.055 with *p* > 0.05 (Sample 2), 1.095 ± 0.064 with *p* > 0.05 (Sample 3), 4.344 ± 0.343 with *p* < 0.001 (LPS), 6.172 ± 0.258 with *p* < 0.001 (Sample 1 + LPS), 6.140 ± 0.276 with *p* < 0.001 (Sample 2 + LPS) and 5.494 ± 0.187 with *p* < 0.01 (Sample 3 + LPS).

Those obtained from the quantification of TNF-α are: 1.002 ± 0.091 (Control), 0.956 ± 0.115 with *p* > 0.05 (Sample 1), 1.132 ± 0.105 with *p* > 0.05 (Sample 2), 1.061 ± 0.087 with *p* > 0.05 (Sample 3), 2.879 ± 0.108 with *p* < 0.001 (LPS), 2.722 ± 0.105 with *p* > 0.05 (Sample 1 + LPS), 3.598 ± 0.159 with *p* < 0.001 (Sample 2 + LPS) and 2.811 ± 0.094 with *p* > 0.05 (Sample 3 + LPS) ([Fig pharmaceuticals-17-01412-ch005]).

While the results on TNF-α are inconclusive in ascertaining the anti-inflammatory action of propolis on immune cells, those on IL-8 highlight its obvious immunomodulatory activity.

The outcomes of the various experiments could provide a deeper understanding of the mechanisms of action of propolis on the wound healing process.

The scratch wound healing assays and the quantifications of the GFs proved that propolis does not have a direct positive influence on healing by promoting cell proliferation or migration. Instead, it was able to act on two of the main phases of the process: (1) the inflammation, by both decreasing the epithelial release of pro-inflammatory cytokines and enhancing the recall of immune cells, hastening the resolution of the phase and thus reducing the risks of chronicization; (2) the remodeling, by both inhibiting the activity of ECM-degrading enzymes and reducing their synthesis, improving the settlement of the new cells and ECM and avoiding the excessive degradation of the matrix.

## 3. Discussion

In this study, for the first time, different samples of Eurasian propolis were analyzed and compared, with the characteristic of having been produced with different methods. In fact, the samples under examination were: an organic and resinous propolis, a fluid industrial propolis and a solid industrial one. The aim of this work was first to evaluate the chemical differences between these samples of propolis. Then, moving onto the experimentation in the biological field, the aim was to gauge the variability of the wound healing property, well established by the literature, based on the composition of the sample material, while also confirming the main mechanisms of action underlying this activity. Moreover, great support in the research was provided by the use of specific software for the determination of new targets, through an in silico methodology only recently introduced in the ethnopharmacological field.

As regards the phytochemical analysis, the obtained data have confirmed that phenolic molecules are one of the main components of propolis. Although propolis composition, including TPC, is heavily influenced by a number of factors (mainly botanic and geographic) which cause a high variability in the quantities of its constituents [18], the measured values are along the lines of those of common Eurasian samples.

As propolis is a resinous substance, it is simple to understand why terpenes constitute the main class of molecules of the mixture, with the lighter ones (mono- and sesqui-terpenes) constituting the essential fraction and the heavier ones (di- and tri-terpenes) the resinous one. The experimentation data are relatable to the percentages that can be found in the literature (~50% resins, ~10% essential oils) [19,20], with only Sample 2 that exhibited an inferior concentration, probably due to the industrial procedures of the removal of waxes, extraction and purification to which it was subjected.

Among the phenolic secondary metabolites of plants that are transferred in propolis, flavonoids represent the majority of them in propolis from temperate zones, such as the Eurasian one, and the obtained data proved this observation for the samples under study.

The quantification by means of HPLC-DAD of some of the marker molecules of Eurasian propolis (pinobanksin, pinocembrin, CAPE, chrysin and galangin) acknowledge the profound influence on the TFC of pinocembrin, chrysin and galangin while confirming the presence of pinobanksin, typical of the Eurasian class together with pinocembrin.

CAPE was detected only in Sample 1. This can be justified by the possible hydrolysis of the phenethyl group from the CAPE molecules induced by the purification and other processes which Samples 2 and 3 were exposed to. Galangin and chrysin have very close retention times (RTs) (~18.2 min and ~18.6 min, respectively, obtained by using external standards), and have therefore been quantified as a single peak. Taking into account the coelution of these two molecules, pinocembrin can be considered the main constituent of the samples under study. This finding is in agreement with the literature, according to which pinocembrin is the principal indicator of Eurasian propolis, followed by galangin and chrysin [18,21].

Shifting to biological experimentation, the results of the DPPH and ORAC assays (scavenging/antioxidant capacity) are correlatable to the TPC (R^2^ = 0.88). Indeed, the samples with higher phenolic content (1 > 3 > 2) are also those with greater scavenging activity and TAC. The DPPH values are also comparable to those that can be found in the literature [21].

The evaluation of the possible cytotoxicity of the in vitro tested concentration of the samples revealed that the propolis under study did not cause any significant disturbance in fibroblasts cultures. Differences in the recorded viability are attributable to the intrinsic variability of the biological model, thus allowing for the conclusion that there are no cytotoxic effects on cells given by the samples.

The wound healing properties of propolis are widely reported in the literature and are one of the main reasons for its common application in the pharmaceutical field. The purpose of the scratch wound-healing assays carried out was to evaluate the possible induction of proliferation and migration by propolis on keratinocytes (HaCaT) and fibroblasts (HFF) pure cell lines. Surprisingly, the experiments led to the conclusion that the samples did not invoke a direct response on the cells to alter the healing process. The treated cells healed the groove normally as did the untreated control ones.

To confirm the results of the scratching assays on the main epithelial cells, the following step was to analyze the possible variation in the synthesis/release of GFs, as fundamental components of the proliferation process, in HFF cultures treated with the propolis under study. The chosen GFs to be quantified were FGF-7 and TGF-β. Specifically, the latter one was determined through LAP, a homodimer relevant in regulating the activity of TGF-β and contained in the synthetized precursor of TGF-β.

The quantification experiments results showed no statistically significant variation in the levels of both FGF-7 and LAP.

According to the data obtained from the scratching and quantification assays, it was possible to conclude that the propolis samples did not have a direct effect on the proliferation capacity of the main epithelial cell lines, pointing to the fact that the wound-healing properties already confirmed by many sources in the literature should be attributed to indirect outcomes of the treatment with propolis in other phases of the healing process.

To test other mechanisms of action of propolis in the healing process and, consequently, to expand the pool of possible targets to put under test, the NP analysis was introduced in the study.

The first software used was SwissTargetPrediction. By inserting the compound of interest, the web-based application provides up to one hundred possible macromolecular targets which should interact with the molecule added as input. Each target is associated with a probability calculated based on the similarity of the molecule with a database of known small molecules that surely interact with that target.

For this study, the molecules analyzed with SwissTargetPrediction were the five representative phytoconstituents of Eurasian propolis (pinobanksin, pinocembrin, CAPE, chrysin and galangin). Among the numerous targets obtained, MMPs were one class which occurred quite frequently as output.

Recognizing it from the literature as one of the key participants in wound healing, MMP9 was chosen as the next objective of the experiments for this study. In the context of an inflammatory and oxidative stress, typical of any common wound, HFF cultures were treated with the samples of propolis for 24 h and then the levels of MMP9 were quantified through an ELISA. All co-treatments induced a statistical reduction in the levels of MMP9.

Having recognized the properties of propolis to induce a decrease in MMP9 expression in fibroblasts, the following test expanded the experimentation to more MMPs. Among the various enzymes which operate on ECM components, collagenases represent one of the main types involved in the restoration of damaged tissues. Their abnormal induction and/or inhibition could significantly alter the finely coordinated mechanism of wound healing.

The assay, based on the evaluation of the capacity of substances to inhibit the activity of different collagenases, was paired with the previous one to establish a global effect of the samples on the ECM-degrading enzymes. Its results show that even low concentrations of propolis are capable of inhibiting the activity of collagenases.

To explore other possible pathways by which propolis could affect the healing process, another software, GeneRecommender, was employed to further analyze the results obtained with SwissTargetPrediction. By adding the target outputs of the latter application, selected based on their connection with wound healing, as input in GeneRecommender, up to fifty new targets (both genes and proteins) closely connected to the query were suggested by its AI engine.

After inserting twenty-four targets derived from the previous analysis of the five marker molecules for Eurasian propolis with SwissTargetPrediction, thirty new targets were proposed by GeneRecommender AI engine as output. Besides different outputs associated with signaling pathways, GFs and their receptors, various targets (e.g., CXCL12, CXCL8, IL6, TNF, PTK2B) were comprised into the inflammatory and/or immunomodulatory fields, a connection well justified by the literature due to the proven anti-inflammatory and immunomodulatory properties of propolis.

Given the firm relationship between the four phases of wound healing, which all contribute to a fast and healthy recovery, the study moved towards the second phase of the healing process, inflammation, heavily influenced by the cytokines and chemokines released by the epithelial cells and by the immunity cells summoned in the site of the injury.

In this regard, tests included the anti-inflammatory activity of the samples on stressed fibroblasts and monocytes and the immunomodulatory properties on the latter ones. Specifically, quantification assays were carried out for IL-1β on HFF and for IL-8 and TNF-α on THP-1.

Even though the production of IL-1β in stressed fibroblasts was not particularly high, Samples 1, 2, and 3 all statistically decreased its synthesis, confirming their anti-inflammatory action on epithelial cells.

In reverse, the same activity in the immunity cells with TNF-α showed inconclusive results.

On the immunomodulatory side, instead, all the sample co-treatments proved to significantly influence the synthesis of IL-8, increasing its levels compared to the sole LPS treatment.

In summary, propolis showed indirect wound-healing properties by mitigating inflammation and remodeling (reduced IL-1β and MMP9) and potentially modulating the immune response (upregulated IL-8). In vitro studies confirmed these effects, demonstrating decreased MMP9 production and collagenase inhibition when cells were co-treated with propolis and a stressor. Propolis also suppressed IL-1β release in fibroblasts, although its impact on TNF-α was inconclusive. Notably, co-treatment upregulated IL-8 in monocytes, suggesting a potential immunomodulatory role. Therefore, we deduced that Eurasian propolis may not directly stimulate cell proliferation during wound healing, while its anti-inflammatory and immunomodulatory properties could indicate an indirect contribution to the process.

## 4. Materials and Methods

The experimental part of this work included the phytochemical characterization of the three Eurasian propolis samples, followed by their in vitro testing using HaCaT, HFF and THP-1 as cell lines, together with cell-free assays, to evaluate the effects on the wound- healing process. All the assays, both the phytochemical and biological analyses, were performed in triplicate.

### 4.1. Samples

“Sample 1” was chosen as a representative of an Italian, Eurasian propolis among thirty-two samples, all of them produced in spring 2020 in various Italian regions and provided by local beekeepers. Major debris and waxes had already been removed when the propolis were supplied. The samples were solubilized in EtOH 80% *v*/*v* to a 100 mg/mL concentration and stocked. For the analysis and experiments carried out, aliquots of Sample 1 were further diluted to a 10 mg/mL concentration with EtOH 80% *v*/*v*.

“Sample 2” is a chemically characterized commercial liquid extract provided by an Italian company and made from Chinese Eurasian propolis. It was the only liquid extract of all the ones tested. It is a EtOH 70% *v*/*v* solution made with 400 mg/mL of raw propolis that was further diluted with EtOH 80% *v*/*v* to a 10 mg/mL concentration.

“Sample 3” is a dry extract supplied by a second Italian company. It was dissolved in EtOH 80% *v*/*v* with an ultrasonic bath (30 min sonication) to obtain a 10 mg/mL concentrated solution.

All samples were filtered with syringe filters with pore size 0.45 μm and 0.2 μm before chemical and biological experiments, respectively.

### 4.2. Phytochemical Analysis

Standard procedures were used to determine both the qualitative and quantitative phytochemical composition of the samples.

#### 4.2.1. Total Phenolic Content

A modified version of the original Folin–Ciocâlteu (FC) colorimetric assay [22] was used to determine the TPC. 20 μL of sample were diluted with distilled water to a final volume of 3 mL. A total of 500 μL of FC reagent (Sigma-Aldrich^®^, Milan, Italy), previously diluted 1:10 in distilled water, was added, and the mixture was gently stirred; then, 1 mL of a 30% *w*/*v* aqueous solution of sodium carbonate was added. Mixtures were kept in the dark at room temperature for 15 min to stabilize the reaction, then poured into cuvettes. Pure ethanol and distilled water were used as blanks. The absorbance of the samples was determined at a wavelength of 760 nm with Shimadzu UV-Vis Spectrophotometer UV-1900i.

The total phenolic content was determined using a calibration curve constructed with gallic acid (Sigma-Aldrich^®^; 0.25–5 mg/mL, R^2^ = 0.998). Results were expressed as % of total phenolics as GAE.

#### 4.2.2. Total Flavonoid Content

The evaluation of the TFC through a modified version of the protocol in Sosa et al. (2007) [23] was based on the absorbance of 250 μL of the diluted samples (1:200 EtOH 80% *v*/*v*) in a 96-wells plate at a wavelength of 353 nm, the maximum absorption wavelength of galangin which was used as standard. EtOH 80% *v*/*v* was used as blank. The absorbance reading was made with PerkinElmer VICTOR^®^ Nivo^TM^ Multimode Microplate Reader (Waltham, MA, USA).

The total flavonoid content was determined using a calibration curve constructed with galangin (0–20 µg/mL, R^2^ = 0.99). Results were expressed as % of total flavonoids as galangin equivalents.

#### 4.2.3. Triterpenoid Content

A modified colorimetric assay based on the reaction of the samples with a vanillin solution in acetic acid and perchloric acid [24] was employed to estimate the triterpenoids concentration. A total of 10 μL of each sample were diluted with 190 μL of their extraction solvent and 300 μL of a 5% *w*/*v* solution of vanillin in glacial acetic acid were added. After stirring the mixture, 1 mL of perchloric acid was added, then the tubes were incubated at 60 °C for 45 min and later left to cool down to room temperature. The solutions then had 3.5 mL of glacial acetic acid added to them, and, after transferring 250 μL of each solution into a 96-wells plate, the absorbance was read at a wavelength of 540 nm with SAFAS Monaco microplate absorbance reader SAFAS MP96.

The triterpenoid content was calculated using a calibration curve constructed with β-sitosterol (Sigma-Aldrich^®^; 0–10 mg/mL, R^2^ > 0.99). Results were expressed as % of terpenoids as β-sitosterol equivalents.

#### 4.2.4. Chemical Characterization by HPLC-DAD

A more in-depth analysis of the phenolic and flavonoid contents of the four samples, focused on the main phytoconstituents of interest (pinobanksin, pinocembrin, CAPE, chrysin and galangin), was carried out via HPLC-DAD using a Shimadzu Prominence-i LC-2030C 3D Plus instrument equipped with a Bondapak^®^ C18 column, 10 µm, 125 Å, 3.9 mm × 300 mm (Waters Corporation, Milford, MA, USA). A mixture of the solutions (A) Formic acid 0.1% *v*/*v* in water and (B) Formic acid 0.1% *v*/*v* in methanol were used as a mobile phase. The analysis method applied was: (B) from 40% at 0.01 min to 65% at 12.00 min, 70% at 25.00 min, 75% at 30.00 min, 85% at 35.00 min, 40% at 39.00 min and stop at 45.00 min. Flow rate was set at 0.75 mL/min. Chromatograms were recorded at 280 nm.

Various quantities (0.5–5 μg) of the chosen standards (100 μg/mL diluted solutions) were used to obtain calibration curves to quantify the associated peaks in each sample (30 and 50 μg, 5 mg/mL diluted solutions). Compounds peaks were identified by comparing their retention times and UV spectra with those of the corresponding standards.

### 4.3. Biological Analysis

Excluding the assays concerning the antioxidant/scavenging activity and the ECM-degrading enzymes (cell-free), the biological tests carried out on the samples evaluated their toxicity and effects on cell lines of interest in the healing process. The final concentration of samples for the assays on cell lines was 10 μg/mL. Incubation of the cells used during the experiments took place in a humidified incubator at 37.0 °C and 5.0% CO_2_.

#### 4.3.1. Scavenging Activity and Total Antioxidant Capacity

The DPPH and the ORAC assays allowed the evaluation of the scavenging and antioxidant activities of the samples, respectively.

In this study, the DPPH assay protocol already published in Bonetti et al. (2021) [25] was employed, and PerkinElmer VICTOR^®^ Nivo^TM^ Multimode Microplate Reader was used to read the absorbance values.

The percentage of DPPH neutralization/inhibition was calculated from the absorbance data of the various concentrations of each sample using the formula *% DPPH inhibition* = [(*A*_0_ − *A*_1_)/*A*_0_] × 100, where *A*_0_ and *A*_1_ are the absorbances of the control and the sample, respectively. The scavenging activity was expressed as IC_50_, determined through linear regression.

The ORAC Antioxidant Capacity Assay Kit (KF01004, BQC Redox Technologies, Oviedo Asturias, Spain) was used to evaluate the TAC of the samples in this study. The experiment followed the protocol provided with the kit. The fluorescence (Ex.: 485 nm/Em.: 528 nm) was read for 30 min in intervals of 3 min using PerkinElmer VICTOR^®^ Nivo^TM^ Multimode Microplate Reader.

The TAC was expressed as TEAC, determined using a calibration curve constructed with the Trolox from the assay kit.

#### 4.3.2. Cell Lines

Aneuploid immortal keratinocytes from adult human skin (HaCaT) and Human Foreskin Fibroblasts (HFF) were cultured in Dulbecco’s Modified Eagle’s Medium (DMEM) High glucose supplemented with 10% *v*/*v* Fetal Bovine Serum (FBS) and 1% *v*/*v* penicillin/streptomycin solution. The media and material for cell cultures were supplied by Merck. Cells were maintained under a humidified atmosphere of 5% CO_2_ at 37 °C.

The human monocytic cell line (THP-1) was cultured using RPMI-1640 medium treated as the DMEM one (10% FBS, 1% penicillin/streptomycin solution). A viable cells count was performed using a hemocytometer after staining with Trypan Blue dye.

#### 4.3.3. Cytotoxicity Assay

The Cell Counting Kit-8 (CCK-8, 96992, Sigma-Aldrich^®^) was used to evaluate the cell viability of HFF in response to the samples. The protocol of the assay was the one supplied with the kit.

Cells were seeded at a density of 5 × 10^3^ cells/well in 96-well plates. After 24 h of incubation at 37 °C in a 5% CO_2_ atmosphere, the medium was replaced with fresh DMEM containing 1% FBS and the designated treatments, including the samples solution (10 μg/mL). The plates were then incubated again at 37 °C in a 5% CO_2_ atmosphere for an additional 24 h. After this period, the medium was removed and replaced with fresh medium containing 10% *v*/*v* CCK-8. The absorbance was measured after ~30 min of incubation with the CCK-8 solution using PerkinElmer VICTOR^®^ Nivo^TM^ Multimode Microplate Reader [26].

#### 4.3.4. Cell Migration and Proliferation Assay (Scratch Wound Healing Assay)

The procedure is a modified version of the one from [27]. In detail, HaCaT and HFF cells were seeded into 6-well plates at a density of 50,000 cells/well with DMEM supplemented with 10% FBS and pre-incubated for 24 h until ~80% confluence as a monolayer. Using a 1 mL pipette tip, the monolayer was scratched to create a cross in each well, then the medium and the detached cells were removed and DMEM supplemented with 3% FBS and the samples (10 μg/mL) were added in the wells. Photos of the crosses in each well were taken 0, 2, 6 and 24 h after treatment by using a Leica DMIL microscope (Leica, Wetzlar, Germany).

The size of the gap was evaluated using IC Measure software (The Imaging Source LLC, Version 2.0.0.286 [28]. Accessed on 10 September 2024) by calculating the average of the diagonals of each cut. Confrontations were made between treated cuts and the healing of non-treated ones. The percentage of wound closure was calculated using the formula *% wound closure* = [(*S*_0_ − *S_t_*)/*S*_0_] × 100, where *S*_0_ and *S_t_* are the areas of the wound at the beginning and after the time *t*, respectively.

#### 4.3.5. Quantifications Through ELISAs

For the assays involving FGF-7 (RAB0188, Sigma-Aldrich^®^) and LAP (TGF-β1) (BMS2065, Thermo Fisher Scientific Inc., Waltham, MA, USA), HFF cells (10,000 cells/well) were seeded into 24-well plates. After 24 h of incubation, the cells were treated with 10 μg/mL of sample.

For the MMP9 (BMS2016-2, Thermo Fisher Scientific Inc.) and IL-1β (88-7261, Thermo Fisher Scientific Inc.) assays, HFF cells (10,000 cells/well) were also seeded into 24-well plates and incubated for 24 h. Subsequently, the cells were co-treated with 10 μg/mL of the sample, along with stimulation by LPS (1 µg/mL) and H_2_O_2_ (1.96 × 10^−4^ M) for 24 h.

In the TNF-α (88-7346-22, Thermo Fisher Scientific Inc.) and IL-8 (88-8086-22, Thermo Fisher Scientific Inc.) assays, cytokine and chemokine quantification was performed using THP-1 cell cultures. The cells were co-treated with 10 μg/mL of the sample, along with LPS (100 ng/mL) for 24 h.

All treated and untreated cells underwent three freeze-thaw cycles at −80 °C/+20 °C. ELISA assays were performed according to the manufacturer’s instructions. A PerkinElmer VICTOR^®^ Nivo^TM^ Multimode Microplate Reader was used for the analysis [29].

#### 4.3.6. Inhibition of ECM-Degrading Enzymes

The Collagenase Activity Colorimetric Assay Kit (MAK293, Sigma-Aldrich^®^) was used to determine the inhibition activity of the samples on these enzymes. Different volumes of each sample, corresponding to specific predetermined concentrations, were tested in this study. The samples were prepared by transferring each volume into two wells and adding 10 µL of provided Collagenase, while the positive control (two wells) was prepared by inserting only 10 µL of Collagenase. The volume in each well was then adjusted to 100 µL with Collagenase Assay Buffer. The protocol was the one provided with the kit. The absorbance was read using a PerkinElmer VICTOR^®^ Nivo^TM^ Multimode Microplate Reader.

The inhibition activity was determined using the differences in absorbance between the last and first measurements for each sample referred to that of the non-treated enzymes. Through regression curves constructed for every sample, an approximate value of IC_50_ was calculated.

### 4.4. Statistical Analysis

The data are expressed as mean ± standard deviation (SD). Statistical analyses were conducted using either the unpaired Student’s t-test or one-way ANOVA, with a significance level of *p* < 0.05. Tukey’s post-hoc test was applied for multiple comparisons where appropriate. All analyses and figures were generated using GraphPad Prism version 10.1 (San Diego, CA, USA).

### 4.5. In Silico Target Prediction

Computational analysis used for prediction and evaluation of possible molecular targets was based on the use of two free platforms available on the web.

SwissTargetPrediction is a platform developed by the Molecular Modeling Group of the SIB. Based on the chemical and physical characteristics of the input and the similarities of the compounds and the data provided, the trained algorithm provides a functionality and relevance score, allowing the combination of query terms and providing the relevant literature, on which the matching is based, also evaluating Tanimoto similarity calculations based on the compound annotations derived from ChEMBL [17,30].

Throughout the research process for this study, SwissTargetPrediction was used by inserting as query molecules the five main phenols of interest in propolis: pinobanksin, pinocembrin, CAPE, galangin and chrysin.

GeneRecommender (TheProphetAI S.r.l. [31]. Accessed on 10 September 2024) is a platform that makes use of a proprietary neural network called DeepProphet and relies on other prediction platforms such as GenaMANIA [32], an interactive and visual online protein interaction prediction tool. GeneMANIA requires a list of gene queries to use available genomics and proteomics data to search for functionally similar genes to predict interacting genes for the target gene [33,34].

In this work, the chosen input genes, obtained from the previous analysis with SwissTargetPrediction, were: FLT4, PIK3CB, FGFR1, MMP2, MAPK1, HIF1A, MMP9, IKBKB, PIK3CG, MET, VEGFA, PDGFRB, PGF, PIK3CA, KDR, PDGFRA, IGF1R, EGFR, KIT, CXCR1, PIK3CD, MMP1, ELANE and TLR9.

## 5. Conclusions

What emerged from this research work on three kinds of Eurasian propolis is that despite the different production method, the phytochemical profile of the samples is comparable, and in the same way the biological activities on human keratinocytes and fibroblasts are also aligned. From a phytochemical point of view, this confirmed the peculiar composition of Eurasian propolis regarding its TPC, TFC and the presence of characteristic marker molecules. From the biological point of view, we showed how the well-known wound healing activity of Eurasian propolis is not related to a direct action as a stimulating agent on the proliferation of the epithelial cells, but it can be traced back to an upstream increased activity of the immune system at the site of the lesion and a direct modulation of the downstream remodeling phase.

Nevertheless, further exploration should focus on the probable indirect stimulus on epithelial proliferation from the action of propolis on immunity and endothelial cells, possible sources of growth factors and other stimulating agents active on keratinocytes and fibroblasts, which are physiologically in constant communication with each other.

## Data Availability

No data were used for the research described in the article.

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
