# Peer review of "Phytochemical Characterization and Assessment of the Wound Healing Properties of Three Eurasian Propolis"

_pharmaceuticals, 2024, doi:10.3390/ph17111412_

Round 1
Reviewer 1 Report
Comments and Suggestions for Authors
1) How was it determined that the propolis used in this study was of the poplar type? Pollen analysis is required to determine whether it is poplar type propolis, but such an analysis is not possible here.By examining the article I suggested, propolis should be named according to the region where it was collected, not according to its botanical characteristics.
2)The original methods used to determine total phenolic and flavonoid content should be referenced. https://doi.org/10.1007/s00217-023-04208-x
3) In addition, after the total phenolic substance amounts of propolis are expressed in mg GAE/g, their properties according to different propolises should be discussed. Why were both gallic acid and galangin equivalents studied in this study? According to the article I suggested, the value found according to galangin equivalent should have been higher.
Comments on the Quality of English LanguageSince my English is not very good, this topic cannot be evaluated.
Reviewer 2 Report
Comments and Suggestions for Authors
In the abstract, it is advisable to remove subsections (objectives, methods, results, conclusion) and to add some number data of your specific research results.
In the introduction the paragraph "Propolis, a mixture of substances gathered by bees from a variety of plant sources..." requires references.
The introduction should include information about the chemical composition of propolis, which are already available with references. Please add this data.
The content of flavonoids in sample 4 is 0.25 ± 0.20%, which is not fully correct. The deviation is over 80%. It is not quite correct to present the results in this form.
It is not quite correct to compare the percentages of the sum of phenolic compounds and flavonoids. There were different methods used and the conclusion that flavonoids compose up to 90% of all phenolic compounds in not fully correct. This can be confirmed using the HPLC method, but not by spectrophotometric methods.
The sample 4 is water-soluble. It’s not fully described how it was prepared, but it significantly distinguishes its phytochemical characteristics from samples 1-3. So, an impression is formed that it may essentially act as a placebo, but in some experiments, it is quite effective. This is not really well discussed in the manuscript. Should the sample 4 be removed from the manuscript? It raises a lot of questions that are not discussed by the authors. It is more correct to compare samples 1-3. Otherwise, it should be discussed more thoroughly.
The conclusions need to be refined. They should be shortened and it is necessary to indicate your specific research results.
Elisabetta Miraldi was mentioned 9 times, Mario Biagi - 10 times and Giulia Bani - 7 times in 30 references. This is slightly above the allowed level of self-citation
The novelty of the authors' research and results should be emphasized and highlighted. What was done for the first time?
Round 2
Reviewer 2 Report
Comments and Suggestions for Authors
No comments